# Midazolam Ameliorates Hyperglycemia-Induced Glomerular Endothelial Dysfunction by Inhibiting Transglutaminase 2 in Diabetes

**DOI:** 10.3390/ijms23020753

**Published:** 2022-01-11

**Authors:** Jae-Ah Seo, Nilofar Danishmalik Sayyed, Yeon-Ju Lee, Hye-Yoon Jeon, Eun-Bin Kim, Seok-Ho Hong, Soyeon Cho, Minsoo Kim, Kwon-Soo Ha

**Affiliations:** 1Department of Molecular and Cellular Biochemistry, School of Medicine, Kangwon National University, Chuncheon 24341, Kangwon-do, Korea; rushcar217@gmail.com (J.-A.S.); sayyednilofar052@gmail.com (N.D.S.); dlduswn147@hanmail.net (Y.-J.L.); skyjhy7412@kangwon.ac.kr (H.-Y.J.); eunbin0704@kangwon.ac.kr (E.-B.K.); 2Department of Internal Medicine, School of Medicine, Kangwon National University, Chuncheon 24341, Kangwon-do, Korea; shhong@kangwon.ac.kr; 3Department of Anesthesiology and Pain Medicine, School of Medicine, Kangwon National University, Chuncheon 24341, Kangwon-do, Korea; ju_nu7@naver.com

**Keywords:** diabetic kidney disease, glomerular endothelial dysfunction, microvascular leakage, midazolam, transglutaminase 2

## Abstract

Midazolam is an anesthetic widely used for anxiolysis and sedation; however, to date, a possible role for midazolam in diabetic kidney disease remains unknown. Here, we investigated the effect of midazolam on hyperglycemia-induced glomerular endothelial dysfunction and elucidated its mechanism of action in kidneys of diabetic mice and human glomerular microvascular endothelial cells (HGECs). We found that, in diabetic mice, subcutaneous midazolam treatment for 6 weeks attenuated hyperglycemia-induced elevation in urine albumin/creatinine ratios. It also ameliorated hyperglycemia-induced adherens junction disruption and subsequent microvascular leakage in glomeruli of diabetic mice. In HGECs, midazolam suppressed high glucose-induced vascular endothelial-cadherin disruption and endothelial cell permeability via inhibition of intracellular Ca^2+^ elevation and subsequent generation of reactive oxygen species (ROS) and transglutaminase 2 (TGase2) activation. Notably, midazolam also suppressed hyperglycemia-induced ROS generation and TGase2 activation in glomeruli of diabetic mice and markedly improved pathological alterations in glomerular ultrastructure in these animals. Analysis of kidneys from diabetic Tgm2^−/−^ mice further revealed that TGase2 played a critical role in microvascular leakage. Overall, our findings indicate that midazolam ameliorates hyperglycemia-induced glomerular endothelial dysfunction by inhibiting ROS-mediated activation of TGase2.

## 1. Introduction

Diabetic nephropathy, a major microvascular complication affecting approximately 40% of patients with diabetes mellitus, is the leading cause of chronic kidney disease, leading to end-stage renal disease [1,2]. Diabetic nephropathy is characterized by pathophysiological changes of the kidney, including albuminuria, a progressive decrease in glomerular filtration rate, glomerulosclerosis and tubulointerstitial fibrosis due to increased expression of extracellular matrix, glomerular and tubular membrane thickening, and vascular dysfunction [3,4]. The prevalence of diabetic nephropathy has risen over the past decades; however, current treatment options remain limited and are only able to prevent or delay disease progression [2,5]. In many patients, diabetic nephropathy can be present for several years before it is diagnosed [2,6]. Thus, there is an urgent need for studies aimed at exploring the pathological mechanisms of diabetic kidney disease (DKD) to facilitate development of more effective therapies.

An alteration in glomerular microvascular permeability may be a key pathogenic factor of DKD and is associated with microalbuminuria [7], which is widely accepted as the first clinical sign of diabetic nephropathy. Microalbuminuria is caused by disruption of the glomerular filtration barrier due to foot process effacement of podocytes, glomerular basement membrane (GBM) thickening, and microvascular dysfunction [3,8]. Notably, microvascular dysfunction plays a critical role in the pathogenesis of diabetic microvascular complications, such as diabetic retinopathy, neuropathy, and pulmonary disease [9,10,11]. In diabetic retinas and lungs, microvascular dysfunction results from hyperglycemia-induced elevation of vascular endothelial growth factor (VEGF) expression and subsequent disruption of vascular integrity and microvascular leakage [9,10]. In the kidney, glomerular endothelial cells engage in crosstalk with podocytes and mesangial cells via various ligands, including VEGF and transforming growth factor-β1 (TGF-β1) [8,12], and glomerular endothelial dysfunction has been implicated in development and progression of DKD [3,7]. However, the underlying mechanisms by which hyperglycemia induces alteration of glomerular endothelial permeability and subsequent microalbuminuria remain unclear.

The benzodiazepine compound midazolam is an anesthetic widely used for anxiolysis and procedural sedation [13]. By binding the γ-aminobutyric acid type A (GABA_A_) receptor, midazolam potentiates inhibition of GABA in the central nervous system (CNS) and induces central respiratory system depression and modest hemodynamic changes [13,14]. This compound also binds the mitochondrial outer membrane protein translocator protein, which functions in cholesterol transport, steroid hormone synthesis, and immunomodulation [15], and can protect primary cortical neuronal cells from oxidative damage and inhibit apoptosis of brain astrocytes [16,17]. Recently, it was reported that midazolam attenuates hyperglycemia-induced microvascular leakage in retinas of diabetic mice [14]. In addition, midazolam can suppress hyperglycemia-induced cancer metastasis by inhibiting VEGF-induced vascular leakage in lungs of diabetic mice [18]. However, to date, it is not known whether midazolam can inhibit hyperglycemia-induced kidney injury.

Here, we hypothesized that midazolam may ameliorate hyperglycemia-induced renal dysfunction by inhibiting glomerular microvascular leakage. We tested this in a diabetic mouse model and found that subcutaneous administration of midazolam alleviated hyperglycemia-induced microalbuminuria and inhibited hyperglycemia-induced adherens junction disruption and microvascular leakage in glomeruli. In human glomerular microvascular endothelial cells (HGECs), midazolam also prevented high glucose-induced VE-cadherin disruption and decreased endothelial cell permeability by inhibiting intracellular Ca^2+^ elevation, leading to suppression of reactive oxygen species (ROS) generation and transglutaminase 2 (TGase2) activation. Furthermore, midazolam treatment normalized hyperglycemia-induced glomerular pathological alterations in kidneys of diabetic mice. Thus, taken together, our findings suggest that midazolam is a possible therapeutic for DKD.

## 2. Results

### 2.1. Midazolam Inhibits Hyperglycemia-Induced Adherens Junction Disruption and Vascular Leakage in Kidneys of Diabetic Mice

To determine whether midazolam can inhibit hyperglycemia-induced kidney injury, we treated diabetic mice with midazolam by subcutaneous injection to the nape of the neck for 6 weeks (Figure 1a). Compared with non-diabetic mice, diabetic mice showed loss of body weight and hyperglycemia, as well as elevated water intake, and these traits were unaffected by midazolam (Figure 1b–d). In contrast, urine albumin/creatinine ratios were significantly higher in diabetic mice than in normal controls, and this was ameliorated by midazolam (Figure 1e). Serum creatinine levels were significantly elevated in diabetic mice and decreased in response to midazolam treatment (data not shown), suggesting that hyperglycemia promotes renal dysfunction in diabetic mice, and this is mitigated by midazolam.

We next tested whether midazolam can inhibit hyperglycemia-induced adherens junction disruption and vascular leakage in kidneys of diabetic mice. Compared to normal mice, we observed decreased VE-cadherin fluorescence intensity, representing adherens junctions, in glomeruli of diabetic mice. Critically, this disruption was normalized by midazolam (Figure 2a,b). We then determined the effect of midazolam on hyperglycemia-induced microvascular leakage, as assessed by extravasation of FITC-dextran. We detected high levels of FITC-dextran extravasation in glomeruli of diabetic mice compared to normal controls, which was suppressed by midazolam (Figure 2c,d). These findings demonstrate that midazolam can prevent renal dysfunction by inhibiting hyperglycemia-induced adherens junction disruption and subsequent microvascular leakage in glomeruli of diabetic mice.

### 2.2. Midazolam Inhibits High Glucose-Induced Elevation of Intracellular Ca^2+^ and Subsequent ROS Generation and TGase2 Activation in HGECs

To investigate the mechanism by which midazolam inhibits hyperglycemia-induced vascular leakage in glomeruli of diabetic mice, we assessed the effect of midazolam on high glucose-induced intracellular events in HGECs. We first determined the effect of midazolam on high glucose-induced changes in intracellular Ca^2+^ levels and found that exposure to high glucose increased intracellular Ca^2+^, whereas this increase was inhibited by midazolam in a concentration-dependent manner, with maximal effect at 20 μmol/L (Figure 3a,b). In contrast, mannitol had no effect on intracellular Ca^2+^ levels. High glucose-induced elevation of intracellular Ca^2+^ was also inhibited by the Ca^2+^ chelator BAPTA-AM, but not by the ROS scavengers N-acetyl cysteine (NAC) and Trolox or the TGase inhibitors monodansylcadaverine (MDC) and cystamine (Figure 3a,c).

We then determined the effects of midazolam on high glucose-induced ROS generation and TGase activation in HGECs. Intracellular ROS levels were elevated in response to high glucose conditions, and that elevation was inhibited by midazolam in a concentration-dependent manner, with maximal effect at 20 μmol/L (Figure 4a). High glucose-induced ROS generation was also suppressed by BAPTA-AM and the ROS scavengers, but not by the TGase inhibitors (Figure 4b), suggesting intracellular Ca^2+^ is involved in high glucose-induced ROS generation. We further found that high glucose increased in situ TGase activity, and this was inhibited by midazolam in a concentration-dependent manner (Figure 4c). High glucose-induced TGase activity was also suppressed by TGase2-specific siRNA, which inhibited TGase2 expression (Figure 4d), but not by control siRNA (Figure 4e), demonstrating that TGase2, but not other members of TGase family, primarily contributes to the high glucose-induced increase in TGase activity in glomerular endothelial cells. High glucose-induced activation of TGase2 was also blocked by BAPTA-AM, the ROS scavengers, and the TGase inhibitors (Figure 4f). However, mannitol had no effect on ROS generation and TGase activity (Figure 4b,f). Thus, taken together, our results show that midazolam inhibits high glucose-induced elevation of intracellular Ca^2+^ levels, thereby preventing high glucose-induced ROS generation and TGase2 activation in HGECs.

### 2.3. Midazolam Inhibits High Glucose-Induced Adherens Junction Disruption, Stress Fiber Formation, and Endothelial Cell Permeability in HGECs

We next examined the effect of midazolam on high glucose-induced adherens junction disruption and stress fiber formation in HGECs by performing VE-cadherin and actin filament staining. We found that high glucose-induced VE-cadherin disruption was attenuated by midazolam treatment (Figure 5a,b), as well as by BAPTA-AM, the ROS scavengers, and the TGase inhibitors. High glucose also activated stress fiber formation, which was prevented by midazolam, BAPTA-AM, Trolox, and MDC (Figure 5c). We then performed glomerular endothelial cell monolayer permeability assays using FITC-dextran to determine the effect of midazolam on high glucose-induced glomerular endothelial cell permeability. Our data indicated that high glucose increased endothelial permeability in vitro, and this was inhibited by midazolam (Figure 5d). High glucose-induced endothelial cell permeability was also inhibited by BAPTA, the ROS scavengers, and the TGase inhibitors (Figure 5d). Our results suggests that midazolam prevents high glucose-induced adherens junction disruption, stress fiber formation, and endothelial cell permeability by inhibiting intracellular Ca^2+^ elevation and subsequent ROS generation and TGase2 activation.

We further studied the role of TGase2 using TGase2-specific siRNA in high glucose-induced adherens junction disruption and endothelial cell permeability in HGECs. High glucose-induced VE-cadherin disruption was prevented by TGase2-specific siRNA transfection, whereas the control siRNA had no effect (Figure 5e). TGase2-specific siRNA also prevented high-glucose-induced endothelial cell permeability (Figure 5f), confirming that TGase2 plays a critical role in high glucose-induced endothelial permeability, and that this likely occurs through VE-cadherin disruption.

### 2.4. Midazolam Inhibits Hyperglycemia-Induced ROS Generation and TGase Activation in Kidneys of Diabetic Mice

To validate our in vitro findings, we treated diabetic mice with midazolam by subcutaneous injection and measured ROS generation and TGase activation in the kidney. Compared to normal mice, hyperglycemia increased ROS levels in glomeruli of diabetic mice, and this effect was suppressed by midazolam treatment (Figure 6a). Average ROS levels in diabetic glomeruli were found to be approximately 1.8-fold higher than in normal glomeruli, and this increase was significantly inhibited by midazolam (Figure 6b). TGase activity in vivo was also increased in the glomerulus and renal cortex of diabetic kidneys compared to normal kidneys, but not in those from diabetic mice treated with midazolam (Figure 6c). Average TGase activity in glomeruli of diabetic mice was found to be about 2.5-fold higher than in normal mice, and this increase was blocked by midazolam (Figure 6d). Thus, our results indicate midazolam suppresses hyperglycemia-induced ROS generation and TGase activation, thereby mitigating adherens junction disruption and microvascular leakage, in glomeruli of diabetic mice.

### 2.5. Midazolam Inhibits Hyperglycemia-Induced Pathological Alterations in Glomerular Ultrastructure and Renal Fibrosis in Kidneys of Diabetic Mice

To further investigate the protective effect of midazolam against hyperglycemia-induced renal injury, we examined hyperglycemia-induced pathological alterations in glomerular ultrastructure and renal fibrosis in kidneys of diabetic mice. TEM analysis revealed marked changes in glomerular ultrastructure in response to hyperglycemia in diabetic mice compared to normal controls, as well as foot process effacement of podocytes and GBM thickening (Figure 7a). Notably, these pathological alterations in glomerular ultrastructure were reversed by midazolam. 

We then assessed the effect of midazolam treatment on hyperglycemia-induced renal fibrosis using Masson’s trichome (blue) and Sirius red staining (red) in diabetic mice (Figure 7b). Our data indicate that hyperglycemia significantly increased collagen deposition, as indicated by Masson’s trichome-positive areas, in kidneys of diabetic mice, and this increase was attenuated by midazolam (Figure 7b,c). A similar inhibitory effect of midazolam on hyperglycemia-induced increase in renal fibrosis was observed by Sirius red staining. These results suggest midazolam ameliorates renal dysfunction by normalizing hyperglycemia-induced pathological alterations in glomerular ultrastructure and microvascular leakage in glomeruli of diabetic mice.

### 2.6. Vascular Leakage and Renal Fibrosis Are Not Observed in Kidneys of Diabetic Tgm2^−/−^ Mice

To determine the role of TGase2 in hyperglycemia-induced vascular leakage and renal fibrosis in the diabetic kidney, we utilized Tgm2^−/−^ mice. In contrast to the high levels of FITC-dextran extravasation observed in kidneys of diabetic wild-type (C57BL/6) mice, we found that hyperglycemia did not induce vascular leakage in kidneys of diabetic Tgm2^−/−^ mice (Figure 8a,b). Further, collagen deposition was not detectable in diabetic Tgm2^−/−^ mouse kidneys by Masson’s trichome staining (Figure 8c,d). Collectively, our results reveal that midazolam ameliorates hyperglycemia-induced glomerular endothelial dysfunction and renal fibrosis by inhibiting hyperglycemia-induced intracellular Ca^2+^ elevation and subsequent ROS generation and TGase2 activation in kidneys of diabetic mice (Figure 8e).

## 3. Discussion

Midazolam is a benzodiazepine medication used for anesthesia, procedural sedation, and trouble with sleeping [13]. Notably, in addition to its role in the CNS, midazolam has several biological effects, including anti-oxidant, anti-inflammatory, anti-apoptotic, and anti-tumor activities, in a variety of cell types [16,17,19,20]. Recently, we demonstrated beneficial effects of midazolam on hyperglycemia-induced microvascular leakage via the GABA_A_ receptors in retinas and lungs of diabetic mice [14,18]. Specifically, we found that intravitreal midazolam injection prevents hyperglycemia-induced ROS generation, TGase activation, and subsequent vascular leakage in diabetic mouse retinas [14]. In lungs of diabetic mice, subcutaneous midazolam treatment suppresses hyperglycemia-induced metastasis of melanoma B16F10 cells by inhibiting VEGF-induced vascular leakage via the GABA_A_ receptors [18]. However, a role for midazolam in DKD has not been shown. In this study, we showed that subcutaneous injection of midazolam mitigated hyperglycemia-induced glomerular endothelial dysfunction and pathological alterations in glomerular ultrastructure in kidneys of diabetic mice. Further, midazolam-treated mice did not show any abnormal behaviors compared with control mice, including breathing, sleeping, food consumption, and water intake. Thus, midazolam may represent a potential therapeutic for treatment of DKD.

A key observation is that midazolam attenuates glomerular endothelial dysfunction and renal fibrosis via inhibition of TGase2 in kidneys of diabetic mice. TGase2 is a member of the TGase family that catalyzes protein cross-linking reactions between glutamine and lysine residues in a Ca^2+^-dependent manner [21]. In addition to its protein cross-linking activity, TGase2 functions as a GTP-binding protein, protein disulfide isomerase, and protein kinase [22,23]. TGase2 is ubiquitously expressed, and its transamidation activity is involved in the pathogenesis of numerous diseases, including cancers, celiac disease, cataracts, and diabetes [9,24,25]. It was previously reported that TGase2 activity and expression are increased in kidneys from diabetic animal models [5,24]; however, the role of TGase2 in pathogenesis of renal endothelial dysfunction in diabetes is not clear. Here, we found that midazolam inhibited high glucose-induced VE-cadherin disruption, stress fiber formation, and endothelial cell permeability by inhibiting TGase2 activation in HGECs. In kidneys of diabetic mice, midazolam treatment inhibited hyperglycemia-induced glomerular microvascular leakage and normalized hyperglycemia-induced pathological alterations and renal fibrosis in glomerular ultrastructure. Critically, glomerular microvascular leakage and renal fibrosis were not detectable in kidneys of diabetic Tgm2^−/−^ mice, demonstrating the important role of TGase2 in hyperglycemia-induced renal dysfunction in diabetes. Previous reports have also implicated TGase2 in hyperglycemia-induced microvascular leakage in both retinas and lungs of diabetic mice [9,10]. Thus, TGase 2 is likely a key enzyme in the pathogenesis of diabetic microvascular complications, including diabetic nephropathy and retinopathy, although it will be necessary to elucidate its role in diabetic neuropathy in future studies.

Endothelial dysfunction may play an important role in DKD pathogenesis; however, the molecular mechanisms by which hyperglycemia induces alterations in glomerular microvascular permeability and microalbuminuria are not clearly understood [3,7]. Microalbuminuria results from injury to the glomerular filtration barrier, which consists of three layers: fenestrated endothelial cells covered with a glycoprotein layer, GBM, and podocytes with foot processes [3]. Glomerulosclerosis, podocyte injury, and GBM thickening are associated with DKD [4,26,27]; however, evidence for the role of glomerular endothelial dysfunction in DKD is limited [1,8,12]. There is known crosstalk between glomerular cells, specifically between glomerular endothelial cells and both podocytes and mesangial cells [8]. VEGF, produced in podocytes, is a major mediator of glomerular endothelial cell–podocyte communication [12]. In the current study, we demonstrated that midazolam treatment ameliorated hyperglycemia-induced endothelial dysfunction as well as foot process effacement of podocytes and GBM thickening in glomeruli of diabetic mice. Thus, midazolam might exert beneficial effects against hyperglycemia-induced pathological alterations in glomerular ultrastructure through preventing endothelial dysfunction in diabetic kidneys, although it is a challenge to elucidate this mechanism.

VEGF plays a key role in pathogenesis of diabetic microvascular complications, including diabetic retinopathy, nephropathy, and pulmonary disease [9,10,12]. Here, we found that high glucose induced intracellular Ca^2+^ elevation and subsequent ROS generation and TGase2 activation, resulting in VE-cadherin disruption and endothelial cell permeability in HGECs. In diabetic mice, hyperglycemia induced ROS generation and TGase2 activation, as well as subsequent adherens junction disruption and microvascular leakage in kidney glomeruli. VEGF elevates intracellular Ca^2+^ levels and increases endothelial cell monolayer permeability in human retinal endothelial cells and human pulmonary microvascular endothelial cells [9,10]. Thus, we speculate that upregulation of VEGF by chronic hyperglycemia induces adherens junction disruption and microvascular leakage via elevation of intracellular Ca^2+^ and ROS levels and the subsequent activation of TGase2 in glomeruli of diabetic mice (Figure 8e).

Diabetic nephropathy affects approximately 40% of diabetic patients and is the leading cause of chronic kidney disease and end-stage renal disease globally. However, current treatments can only slow the progression of kidney damage and control related complications [2,5]. Intensive investigations in animal models have focused on developing drugs to treat podocyte injury and renal fibrosis [5,27,28,29,30,31]. A number of potential anti-fibrotic drugs have been identified in preclinical studies, including neutralizing antibodies against TGF-β1 and connective tissue growth factor and a small molecule inhibitor of apoptosis signal-regulating kinase 1; however, their effects on diabetic nephropathy have been limited [29,30,31]. Here, we demonstrated the important role of TGase2 in hyperglycemia-induced microvascular leakage in the glomerulus, and thus TGase2 inhibitors can be considered as potential therapeutics, although further studies on their pharmacokinetics and efficacy are required.

In this study, we show that subcutaneous administration of midazolam ameliorates hyperglycemia-induced renal dysfunction by inhibiting glomerular microvascular leakage. Midazolam has several advantages as a possible drug for DKD. First, midazolam ameliorates both glomerular endothelial dysfunction and pathological alterations in glomerular ultrastructure induced by hyperglycemia, whereas many potential drugs target only podocyte injury or renal fibrosis [27,28,29,31]. Second, midazolam has been widely used for anxiolysis and sedation in humans, and thus, the length of time required for clinical trials might be relatively short. Third, midazolam is available as a generic medication and is not expensive. Additionally, midazolam is a water-soluble benzodiazepine and can be administered by various routes: intravenous, intramuscular, buccal, intranasal, and subcutaneous. However, the pharmacokinetics of midazolam are affected by obesity, age, hepatic cirrhosis, and heart failure [32,33]. Thus, when applying midazolam for the treatment of DKD, individual dose adjustment is required to avoid excessive accumulation of the drug, which may result in serious adverse effects. Moreover, it is necessary to investigate the duration and efficacy of midazolam in patients with diabetic nephropathy.

In conclusion, midazolam displays beneficial effects against hyperglycemia-induced kidney injury and provide evidence for its mechanism of action in vitro and in diabetic mice in vivo. Critically, midazolam treatment ameliorated hyperglycemia-induced glomerular microvascular endothelial dysfunction by inhibiting intracellular Ca^2+^ elevation and subsequent ROS generation and TGase 2 activation. Thus, this compound may represent a potential therapeutic for treatment of DKD.

## 4. Materials and Methods

### 4.1. Diabetic Mouse Models and Midazolam Treatment

Six-week-old male C57BL/6 mice were obtained from Daehan Biolink (EumSeong, Korea). TGase2-null (Tgm2^−/−^) mice (C57BL/6) [9] were kindly provided by Dr. Soo-Yul Kim (National Cancer Center, Goyang, Korea). All animal experiments conformed to the Guide for the Care and Use of Laboratory Animals from the National Institutes of Health and were approved by the Institutional Animal Care and Use Ethics Committee of Kangwon National University. Mice were maintained under pathogen-free conditions in a temperature-controlled room with a 12-h light/dark cycle. Diabetic mice were generated by a single daily intraperitoneal injection for 5 consecutive days of streptozotocin (50 mg/kg body weight; MilliporeSigma, Burlington, MA, USA) [34]. Mice with fasting blood glucose levels ≥19 mmol/L, polyuria, and glucosuria were considered diabetic.

Six weeks after streptozotocin injections, diabetic mice (*n* = 8) were treated with 10 mg/kg midazolam (Bukwang, Ansan, Korea) by subcutaneous injection to the nape of the neck every 3 days for 6 weeks (Figure 1a). Albumin and creatinine concentrations were determined using mouse albumin (Abcam, Cambridge, UK) and creatine (BioVision, Milpitas, CA, USA) ELISA kits.

### 4.2. Measurement of Vascular Leakage in Mouse Kidneys

Microvascular leakage in the glomerulus of mouse kidneys was measured, as previously described [10]. Briefly, mice were anesthetized with 3% isoflurane, and 1.25 mg of 500-kDa FITC-dextran (MilliporeSigma) was injected into the left ventricle of each mouse and allowed to circulate for 5 min. The kidneys were fixed with 4% paraformaldehyde (MilliporeSigma) and embedded in Optimal Cutting Temperature (OCT) compound (Sakura Finetek, Torrance, CA, USA). Kidney cryosections (20-μm thick) were prepared using a microtome-cryostat (Leica Biosystems, Wetzlar, Germany) and observed by confocal microscopy (K1-Fluo). Microvascular leakage (*n* = 8) was quantitated by measuring the fluorescence intensities of FITC-dextran extravasated from the renal glomerulus.

### 4.3. Visualization of VE-Cadherin in Mouse Kidneys

Kidney tissues were embedded in OCT compound and unfixed cryosections (20-μm thick) were prepared using a microtome-cryostat (Leica Biosystems, Wetzlar, Germany) and incubated overnight with monoclonal VE-cadherin antibody (1:200; BD Pharmingen, San Diego, CA, USA) at 4 °C. Sections were probed with a FITC-conjugated goat anti-mouse antibody (1:300; MilliporeSigma) for 1 h at room temperature and stained with 1 μg/mL 4′,6-diamidino-2-phenylindole (DAPI) for 10 min. VE-cadherin in the glomerulus of kidney tissues (*n* = 8) was visualized by confocal microscopy and quantitated by measuring fluorescence intensity.

### 4.4. Measurement of ROS Generation in Mouse Kidneys

ROS levels in the glomerulus of mouse kidneys were measured by dihydroethidium staining (Thermo Fisher Scientific, Waltham, MA, USA) [10]. Briefly, unfixed cryosections (20-μm thick) were incubated with 10 μmol/L dihydroethidium for 30 min at 37 °C and observed by confocal microscopy (*n* = 8). ROS levels were quantified by measuring fluorescence intensities in the renal glomerulus

### 4.5. Measurement of In Vivo TGase Activity in Mouse Kidneys

TGase transamidation activity in the glomerulus of mouse kidneys was determined by confocal microscopy [18]. Briefly, unfixed cryosections (20-μm thick) were incubated with 1 mmol/L 5-(biotinamido)pentylamine for 1 h, fixed with 4% paraformaldehyde for 30 min, and permeabilized with 0.2% Triton X-100 for 30 min. Sections were incubated with FITC-conjugated streptavidin (1:200, *v*/*v*) for 1 h and 1 μg/mL DAPI for 10 min and observed by confocal microscopy (*n* = 8). TGase activities were quantified by measuring fluorescence intensities in the renal glomerulus.

### 4.6. Renal Histopathological Analysis

Kidney tissues were fixed with 4% paraformaldehyde and embedded in paraffin. Kidney sections (5-μm thick) were prepared using a microtome-cryostat (Leica Biosystems, Wetzlar, Germany), deparaffinized, and stained with Masson’s trichrome or Sirius red to assess collagen deposition (*n* = 8). Fibrosis levels were determined by calculating areas positive for Masson’s trichome staining.

For electron microscopic imaging of mouse glomeruli, kidney tissues were immediately fixed with 2% glutaraldehyde and 2% paraformaldehyde for 1 h at 4 °C and then post-fixed in osmium tetroxide for 40 min at 4 °C. Kidney samples were then dehydrated in a graded series of ethanol and treated with a graded propylene oxide series. Tissues were embedded in EPON resin, and ultrathin sections (80 nm thicknesses) were stained with uranyl acetate and lead citrate. Stained sections were analyzed using TEM (JEM-2100F, Tokyo, Japan) at an accelerating voltage of 200 kV at the Korean Basic Science Institute (Chuncheon, Korea).

### 4.7. Cell Culture

HGECs were purchased from the Applied Cell Biology Research Institute (Cell Systems, Kirkland, WA, USA), and 7–10 passage subconfluent cells were used in all experiments. Cells were grown on 2% gelatin-coated plates in M199 medium, supplemented with 20% fetal bovine serum (FBS), 3 ng/mL basic fibroblast growth factor (bFGF), 5 U/mL heparin, 100 U/mL penicillin, and 100 μg/mL streptomycin, in a humidified 5% CO_2_ incubator. For all experiments, cells were incubated for 12 h in low-serum M199 medium, supplemented with 5% FBS, 0.5 ng/mL bFGF, and 100 μg/mL streptomycin. Cells were then treated with 5.5 mmol/L D-glucose (normal glucose) or 30 mmol/L D-glucose (high glucose) for 3 days; 30 mmol L-mannitol was used as an osmotic control for high glucose treatment.

### 4.8. Measurement of Intracellular Ca^2+^ and ROS Levels

Intracellular Ca^2+^ and ROS levels were monitored by confocal microscopy (K1-Fluo; Nanoscope Systems, Daejeon, Korea), as previously described [24,34]. Cells on coverslips were labeled with 2 μmol/L Flou-4 AM (Thermo Fisher Scientific, Waltham, MA, USA) for 30 min or 10 μmol/L 2′,7′-dichlorodihydrofluorescein diacetate (Thermo Fisher Scientific) for 10 min. Single-cell fluorescence intensities were determined for 30 randomly selected cells from three microscopic fields per experiment. Intracellular Ca^2+^ and ROS levels were determined by comparing fluorescence intensities of treated cells with those of control cells (fold difference).

### 4.9. Measurement of In Situ TGase Activity

In situ TGase transamidation activity was measured by confocal microscopy, as previously described [34]. Briefly, HGECs were incubated with 1 mmol/L 5-(biotinamido) pentylamine for 1 h, fixed in 3.7% formaldehyde for 30 min, and permeabilized with 0.2% Triton X-100 for 30 min. Cells were then incubated with fluorescein isothiocyanate (FITC)-conjugated streptavidin (1:200; MilliporeSigma, Burlington, MA, USA), and fluorescence intensities of single stained cells were determined for 30 randomly selected cells from three microscopic fields per experiment. The contribution of TGase2 to the TGase activation was studied by transfecting endothelial cells with 100 nmol/L TGase2 siRNA or control siRNA (Santa Cruz Biotechnology; Dallas, TX, USA) using siLentFect lipid reagent (Bio-Rad Laboratories; Hercules, CA, USA) according to the manufacturer’s instructions [34].

### 4.10. Visualization of VE-Cadherin and Actin Filaments in HGECs

VE-cadherin was visualized as previously described [10]. Briefly, HGECs were fixed in 3.7% formaldehyde, permeabilized with 0.2% Triton X-100, and incubated with monoclonal VE-cadherin antibody (1:200; Santa Cruz Biotechnology, Dallas, TX, USA). Cells were then probed with a FITC-conjugated goat anti-mouse antibody (1:200; MilliporeSigma), and VE-cadherin was visualized using confocal microscopy (K1-Fluo). Adherens junctions were quantified by measuring peak fluorescence intensities of VE-cadherin histograms at the single-cell level.

Actin filaments were visualized using Alexa Fluor 488 phalloidin (1:200; Thermo Fisher Scientific) as previously described [11].

### 4.11. In Vitro Glomerular Endothelial Cell Monolayer Permeability Assay

Glomerular endothelial cell monolayer permeability in vitro was assessed, as previously described [10]. Briefly, HGECs were grown to confluence on gelatin-coated inserts (0.4 μm, Costar; Corning, NY, USA) and incubated at 37 °C for 3 days with low glucose or high glucose conditions. Treated cells were probed with 1 mg/mL 40-kDa FITC-dextran (MilliporeSigma) for the last 60 min, and the amount of FITC-dextran in the lower chamber was measured with well-type arrays by confocal microscopy [35].

### 4.12. Statistical Analysis

Data were analyzed using OriginPro 2015 software (OriginLab, Northampton, MA, USA). Data are expressed as the mean ± standard deviation (SD) from at least three independent experiments. Statistical significance was determined using one-way ANOVA with Holm–Sidak’s multiple comparisons test, with *p*-values < 0.05 considered statistically significant.

## Figures and Tables

**Figure 1 ijms-23-00753-f001:**
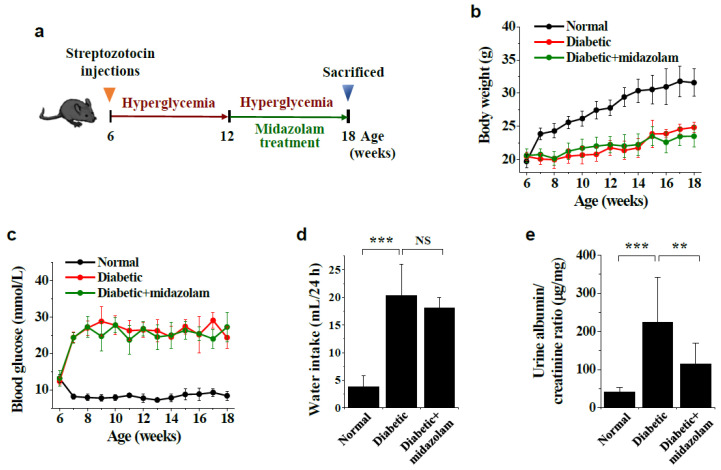
Biochemical parameters of normal, diabetic, and midazolam-treated diabetic mice. Diabetes was induced in 6-week-old male C57BL/6 mice by intraperitoneal injection of streptozotocin (50 mg/kg body weight) for 5 consecutive days. Six weeks after streptozotocin injection, diabetic mice were treated with midazolam (10 mg/kg per mouse) by subcutaneous injection to the nape of the neck every 3 days for 6 weeks. (**a**): Schematic overview of diabetic mouse model and midazolam treatment regimen. (**b**) and (**c**): Body weight (**b**) and blood glucose levels (**c**) were monitored weekly (*n* = 8). (**d**) and (**e**): After 6 weeks of midazolam treatment, water intake (mL/24 h) (**d**) and urine albumin/creatinine ratios (**e**) were measured (*n* = 8). ** *p* < 0.01, *** *p* < 0.001; NS, non-significant.

**Figure 2 ijms-23-00753-f002:**
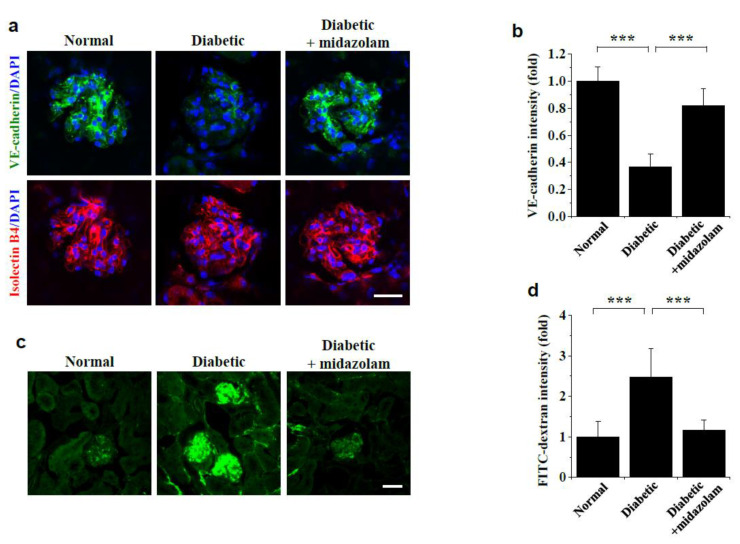
Midazolam treatment inhibits hyperglycemia-induced VE-cadherin disruption and microvascular leakage in kidneys of diabetic mice. Diabetic mice were treated with midazolam via subcutaneous injection for 6 weeks, and VE-cadherin (**a**,**b**) and microvascular leakage (**c**,**d**) were visualized and analyzed by confocal microscopy in the kidneys of normal, diabetic, and midazolam-treated diabetic mice (*n* = 8). (**a**): Representative fluorescence images of VE-cadherin staining, with 4′,6-diamidino-2-phenylindole (DAPI) nuclear counterstaining (blue), in kidney glomeruli. Scale bar, 25 μm. (**b**): Adherens junctions were quantified by measuring fluorescence intensities of VE-cadherin in the renal glomerulus. (**c**): Representative fluorescence images of fluorescein isothiocyanate (FITC)-dextran extravasation in kidney glomeruli. Scale bar, 50 μm. (**d**): Microvascular leakage was quantified by measuring fluorescence intensities of FITC-dextran in the renal glomerulus. *** *p* < 0.001.

**Figure 3 ijms-23-00753-f003:**
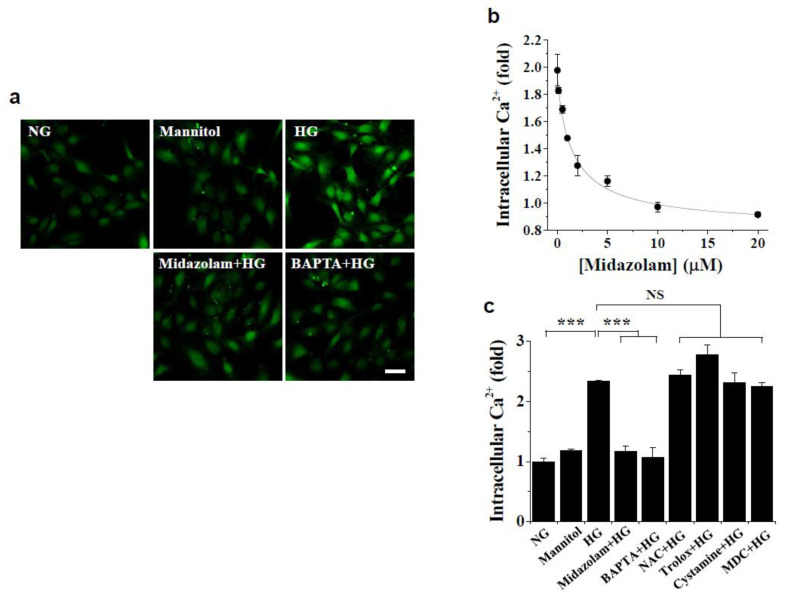
Effects of midazolam and inhibitors on high glucose-induced increase in intracellular Ca^2+^ levels in human glomerular microvascular endothelial cells (HGECs). HGECs were treated for 3 days with normal glucose (NG) or high glucose (HG) at the indicated concentrations of midazolam (MDZ) or in the presence of 20 μmol/L midazolam, 5 μmol/L BAPTA-AM, 1 mmol/L N-acetyl cysteine (NAC), 0.5 μmol/L Trolox, 50 μmol/L cystamine, or 20 μmol/L monodansylcadaverine (MDC). Mannitol was used as an osmotic control. Intracellular Ca^2+^ levels were visualized by confocal microscopy (**a**) and quantified by measuring fluorescence intensities (**b**,**c**). (**a**): Representative images of intracellular Ca^2+^ levels. Scale bar, 50 μm. (**b**): Midazolam inhibits high glucose-induced elevation of intracellular Ca^2+^ levels in a concentration-dependent manner. (**c**): Effect of various inhibitors on high glucose-induced intracellular Ca^2+^ elevation. Results are expressed as the mean ± standard deviation (SD) from three independent experiments. *** *p* < 0.001; NS, non-significant.

**Figure 4 ijms-23-00753-f004:**
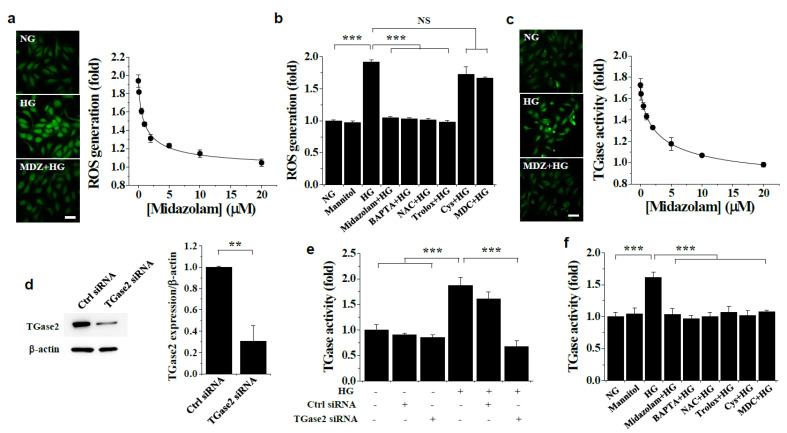
Effects of midazolam and inhibitors on high glucose-induced generation of reactive oxygen species (ROS) and TGase2 activation in HGECs. HGECs were treated for 3 days with normal glucose (NG) or high glucose (HG) at the indicated concentrations of midazolam (MDZ) or in the presence of 20 μmol/L MDZ, 5 μmol/L BAPTA-AM, 1 mmol/L NAC, 0.5 μmol/L Trolox, 50 μmol/L cystamine, 20 μmol/L MDC, or 100 nM TGase2-specific small interfering (si) RNA or control (Ctrl) siRNA. Mannitol was used as an osmotic control. Intracellular ROS levels (**a**,**b**) and in situ TGase transamidation activity (**c**,**e**,**f**) were visualized by confocal microscopy and quantified by measuring fluorescence intensity. Scale bar, 50 μm. (**d**): TGase2-specific siRNA inhibits TGase2 expression. Results are expressed as the mean ± SD from three independent experiments. ** *p* < 0.01, *** *p* < 0.001; NS, non-significant.

**Figure 5 ijms-23-00753-f005:**
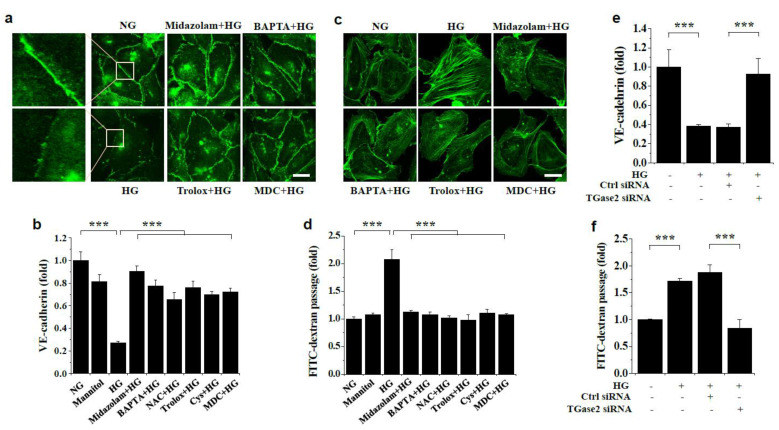
Effects of midazolam and inhibitors on high glucose-induced VE-cadherin disruption, stress fiber formation, and endothelial cell monolayer permeability in HGECs. HGECs were treated for 3 days with normal glucose (NG) or high glucose (HG) in the presence of 20 μmol/L midazolam, 5 μmol/L BAPTA-AM, 1 mmol/L NAC, 0.5 μmol/L Trolox, 50 μmol/L cystamine, or 20 μmol/L MDC. Mannitol was used as an osmotic control. (**a**,**b**): VE-cadherin was visualized by confocal microscopy (**a**) and quantified by measuring fluorescence intensities (**b**). Scale bar, 25 μm. (**c**): Stress fibers were stained using Alexa 488-conjugated phalloidin. Scale bar, 25 μm. (**d**): Glomerular endothelial cell monolayer permeability in vitro was assessed using FITC-dextran. (**e**,**f**): HGECs were transfected with 100 nM TGase2-specific siRNA or control (Ctrl) siRNA and treated with high glucose for 3 days. TGase2-specific siRNA inhibited the high glucose-induced VE-cadherin disruption (**e**) and in vitro endothelial cell permeability (**f**). Results are expressed as the mean ± SD from three independent experiments. *** *p* < 0.001; NS, non-significant.

**Figure 6 ijms-23-00753-f006:**
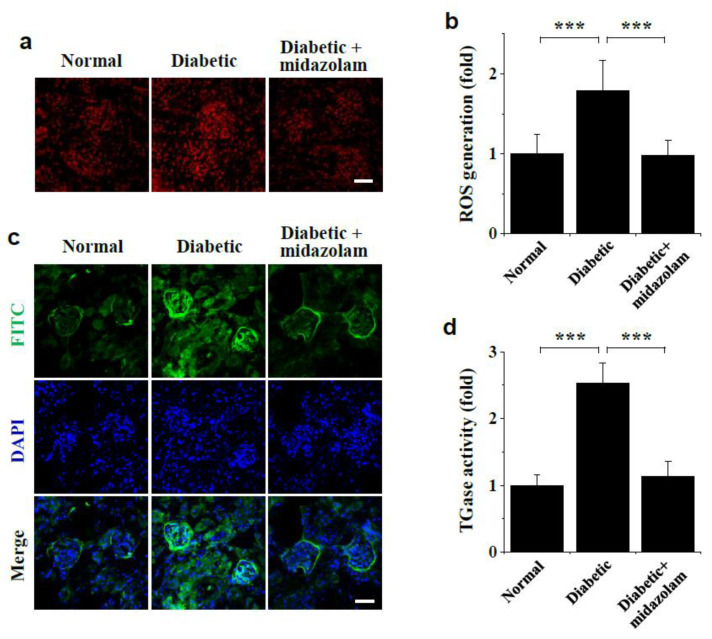
Midazolam treatment inhibits hyperglycemia-induced ROS generation and TGase activation in kidneys of diabetic mice. Diabetic mice were treated with midazolam by subcutaneous injection for 6 weeks, and both ROS generation (**a**,**b**) and in vivo TGase activity (**c**,**d**) were visualized and analyzed by confocal microscopy in kidneys from normal, diabetic, and midazolam-treated diabetic mice (*n* = 8). (**a**): Representative fluorescence images to measure ROS generation in kidney glomeruli. Scale bar, 50 μm. (**b**): ROS levels were quantified by measuring fluorescence intensity in the renal glomerulus. (**c**): Representative fluorescence images of in vivo TGase activity (green) with DAPI (blue) nuclear counterstaining in kidney glomeruli. Scale bar, 50 μm. (**d**): TGase activity was quantified by measuring fluorescence intensity in the renal glomerulus. Scale bar, 50 μm. *** *p* < 0.001.

**Figure 7 ijms-23-00753-f007:**
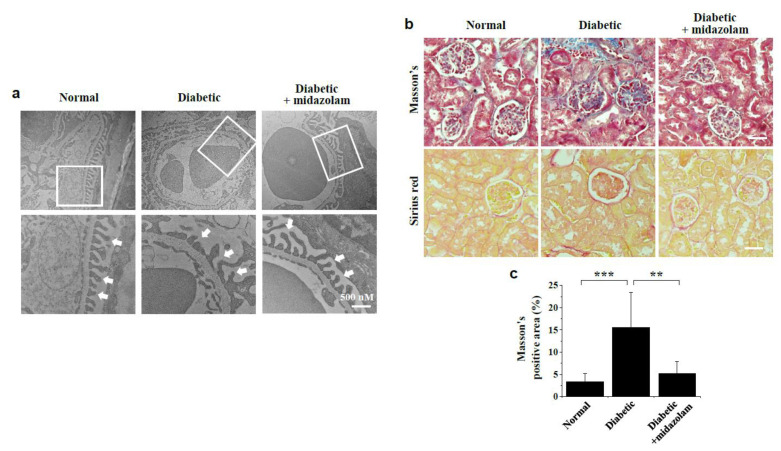
Midazolam treatment inhibits hyperglycemia-induced pathological alterations of glomerular ultrastructure and renal fibrosis in kidneys of diabetic mice. Glomerular ultrastructure was evaluated by transmission electron microscopy (TEM) analysis (**a**), and renal fibrosis (**b**,**c**) was assessed by Masson’s trichrome and Sirius red staining of kidneys from normal, diabetic, and midazolam-treated diabetic mice. (**a**): Representative micrographs of glomerular ultrastructure. The bottom micrographs display the magnified image of the square in the top images and show ultrastructural alterations in podocyte foot processes in the renal glomerulus of diabetic mice (indicated by arrows). Scale bar, 500 nm. (**b**): Representative images of renal tissues stained with Masson’s trichrome and Sirius red. Scale bar, 50 μm. (**c**): Quantification of areas positive for Masson’s trichome staining (*n* = 8). ** *p* < 0.01, *** *p* < 0.001.

**Figure 8 ijms-23-00753-f008:**
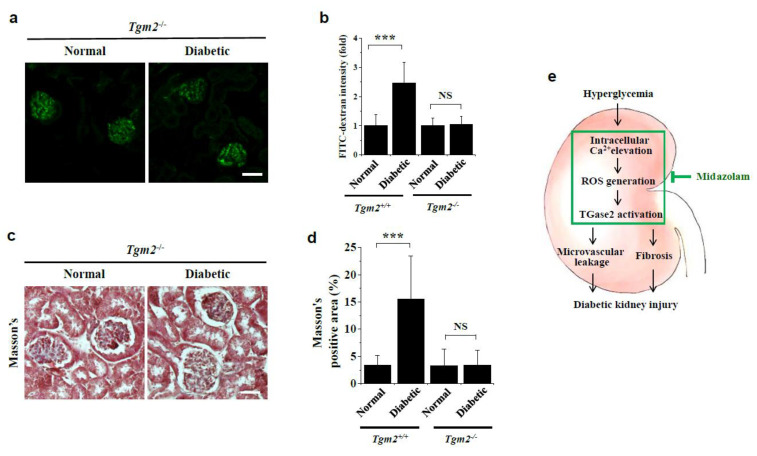
Analysis of vascular leakage and renal fibrosis in kidneys of diabetic Tgm2^−/−^ mice and a schematic diagram depicting the effect of midazolam on the kidney injury in diabetic mice. (**a**,**b**): Vascular leakage of FITC-dextran is not observed in kidneys of normal or diabetic Tgm2^−/−^ mice (*n* = 6). Representative fluorescence images of FITC-dextran are shown (**a**). Scale bar, 50 μm. Vascular leakage was quantified by measuring the fluorescence intensity of FITC-dextran in the renal glomerulus (**b**); Tgm2^+/+^ data are from Figure 2d. (**c**,**d**): Renal fibrosis is not detected in kidneys of normal and diabetic Tgm2^−/−^ mice, as assessed by Masson’s trichrome staining (n = 6). Representative images of renal tissues stained with Masson’s trichrome are shown (**c**). Scale bar, 50 μm. Areas positive for Masson’s trichome staining (**d**); Tgm2^+/+^ data are from Figure 7c. *** *p* < 0.001; NS, non-significant. (**e**): Schematic model depicting the beneficial effect of midazolam on hyperglycemia-induced kidney injury in diabetic mice.

## Data Availability

The data presented in this study are available on request from the corresponding author.

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
