# Peer review of "Midazolam Ameliorates Hyperglycemia-Induced Glomerular Endothelial Dysfunction by Inhibiting Transglutaminase 2 in Diabetes"

_ijms, 2022, doi:10.3390/ijms23020753_

Round 1

Reviewer 1 Report

the authors described in an in vivo diabetic nephropathy model of streptozotocin-treated mice that midazolam is able to improve vascular leakege, maintaining VE-cadherin expression and reducing ROS expression and TGase2 activity . In vitro, they confirm that midazolam act on HGECs by mantaining VE-cadherin expression and inhinibting intracellular Ca2+ elevation and subsequent generation of ROS and TGase2 activity.

This manuscript is very interesting but many questions remain open. Major revisions:

1) on which receptors does midazolam act on HGEC?

2)The authors see in vivo alterations at the level of the podocytes and at the level of the basement membrane. What effect is medazolam mediated on podocytes and mesangial cells?

3)Have any dose-response experiments in vivo been performed with various doses of midazolam? what appears to be the minimum effective dose?

Minor revisions:

1) In the methods, the animal model should be described first and the in vitro model secondarily, in the order of how the results are shown

Author Response

[Reviewer #1]

[Major revisions]

1. On which receptors does midazolam act on HGEC?

â–º In previous reports (references 14 and 18), we reported beneficial effects of midazolam against hyperglycemia-induced microvascular leakage via the GABAA receptors in human retinal endothelial and human pulmonary microvascular endothelial cells. Thus, we discussed these with including the GABAA receptors at the 1st paragraph of p11. Thus, we think the GABAA receptors may involve in the beneficial effects of midazolam on human glomerular microvascular endothelial cells, although we need further studies for elucidating the receptors in these cells.      

2. The authors see in vivo alterations at the level of the podocytes and at the level of the basement membrane. What effect is midazolam mediated on podocytes and mesangial cells?

â–º According to the comments, we discussed this with additional discussion from the 2nd paragraph of p12 to the 1st paragraph of p13.  

3. Have any dose-response experiments in vivo been performed with various doses of midazolam? what appears to be the minimum effective dose?

â–º To determine the optimal dose of midazolam in mice, initially we performed Miles vascular permeability assay after subcutaneously injecting 1, 2, 5, and 10 mg/kg midazolam into the nape of the neck in normal mice. We found that 5 - 10 mg/kg midazolam attenuated VEGF-induced vascular leakage. Then, we found that, in diabetic mice, 10 mg/kg midazolam is optimal in hyperglycemia-induced vascular leakage.

[Minor revisions]

1. In the methods, the animal model should be described first and the in vitro model secondarily, in the order of how the results are shown

â–º According to the comment, we re-ordered the ‘4. Materials and Methods’ section.

Reviewer 2 Report

Seo et al. explored the effect of midazolam on hyperglycemia-induced glomerular endothelial dysfunction and its mechanisms of action in kidneys of diabetic mice and human glomerular microvascular endothelial cells. They found that midazolam administered subcutaneously for 6 weeks ameliorated albuminuria via inhibition of intracellular Ca2+ elevation and subsequent generation of reactive oxygen species (ROS) and transglutaminase 2 (TGase2) activation.

I offer the following suggestions for consideration:

  1. It is not clear from the results how persistent was the experimental effect of midazolam. Please indicate whether and for how long the anti-albuminuric effect persisted after cessation of drug administration, i.e., if there is a structural effect on the glomeruli.
  2. Midazolam is metabolized in the liver and excreted in the urine, which means that midazolam accumulates with declining kidney function or when heart failure is present and can cause prolonged effects. Please discuss these aspects in the paper.
  3. Since the treatment is ultimately intended for usage in humans, the authors should consider discussing the likelihood of midazolam as a therapeutic option for treatment of DKD, including potential administration routes in humans.

Author Response

[Reviewer #2]

1. It is not clear from the results how persistent was the experimental effect of midazolam. Please indicate whether and for how long the anti-albuminuric effect persisted after cessation of drug administration, i.e., if there is a structural effect on the glomeruli.

â–º Preventive effect of midazolam on hyperglycemia-induced vascular leakage persisted for three days. That is why we subcutaneously injected midazolam every three days (see the section 4.1 in the revised manuscript). We measured urine albumin once after 6 weeks of midazolam treatment, and thus we do not have any information how long the anti-albuminuric effect persists.

As for the structural effect, midazolam attenuated hyperglycemia-induced pathological alterations in glomerular ultrastructure. Please see Figure 7 and the section 2.4 (P8-p9).      

2. Midazolam is metabolized in the liver and excreted in the urine, which means that midazolam accumulates with declining kidney function or when heart failure is present and can cause prolonged effects. Please discuss these aspects in the paper.

â–º According to the comment, we discussed this issue at the 2nd paragraph of p14.

3. Since the treatment is ultimately intended for usage in humans, the authors should consider discussing the likelihood of midazolam as a therapeutic option for treatment of DKD, including potential administration routes in humans.

â–º According to the comment, we discussed the possible use of midazolam for the treatment of DKD at the 2nd paragraph of p14.

Reviewer 3 Report

In this manuscript, Seo et al. show that midozolam, a widely used anesthesic, could play a role for protection against hyperglycemia-induced glomerular endothelial dysfunction. They speculated that this effect is dependent to Transglutaminase 2 (TGase2) activation by ROS.

This topic is quite interesting but there are many concerns about the study.

Fig 1:

- Blood urea nitrogen level must be added in the mouse kidney phenotype

Fig 2:

- Authors must provide co-satining of VE-Cadh and an endothelial marker as CD31

Fig4:

- in panel e, 1) the TGase activity inhibition by siRNA is very weak. How the authors explain this discrepancy ?

                     2) Authors didn't precise if HG condition is compared to mannitol or NG?

- Authors must show the consequences of TGase2 inhibition by siRNA on : i) VE-cadh staining, ii) ROS generation, iii) FITC-dextran

Fig5:

- A bigger magnification of VE-cadh staining is necessary to appreciate VE-cadh disruption

Fig6 and Fig7:

- HE staining and collagen 4A staining could improve diabetic mouse phenotypage and the effect of midazolam.

- TEM images are very nice but weaker magnification is needed to appreciate the glomerulus as a whole and to visualize endothelial cells

Discussion:

- It has been shown that the TGase2 is involved in autophagy dependent clearance of ubiquitinated proteins. Moreover emerging data suggest that targeting autophagy is renoprotective during diabetic nephropathy. Thus, it will be interstesting to demontstrate in this model if autophagy patway is modulated by TGase2 activation.

- the only marker used by authors to study adherns junctions is VE-cadh. however, it is necessary to complete this by ZO-1 staining, CD31 (PECAM1) staining and actin cytoskeleton.

Author Response

Response to the 3rd reviewer’s comments (R1)

Ms. Ref. No.: IJMS-1478301R1

Title: Midazolam ameliorates hyperglycemia-induced glomerular endothelial dysfunction by inhibiting transglutaminase 2 in diabetes

 [Reviewer #3]

1. Fig 1:

1-1) Blood urea nitrogen level must be added in the mouse kidney phenotype.

â–ºAccording to the comment, we measured the levels of serum creatinine instead of blood urea nitrogen and described in the Results (p6).

2. Fig 2:

2-1) Authors must provide co-satining of VE-Cadh and an endothelial marker as CD31.

â–º According to the comment, we stained VE-cadherin with Alexa 647-isolectin B4 to stain blood vessels. The results are displayed in Fig. 2a.

3. Fig 4:

3-1) in panel e, 1) the TGase activity inhibition by siRNA is very weak. How the authors

explain this discrepancy?

â–º We think that 70% inhibition of TGase2 expression using TGase2 siRNA is enough to see the role of TGase2 in cells. 

3-2) Authors didn't precise if HG condition is compared to mannitol or NG?

â–º In Fig.4, we include mannitol or normal glucose (NG).

3-3) Authors must show the consequences of TGase2 inhibition by siRNA on : i) VE-cadh

staining, ii) ROS generation, iii) FITC-dextran

â–º ROS generation is not affected by TGase2 activation in endothelial cells. We reported that TGase2 siRNA inhibits high glucose-induced VE-cadherin disassembly and endothelial cell monolayer permeability in two human endothelial cells; human retinal endothelial and pulmonary microvascular endothelial cells (references 9 and 10). We discussed this issue in the Discussion (the 2nd paragraph of 11 to the 1st paragraph of 12).     

4. Fig 5:

4-1) A bigger magnification of VE-cadh staining is necessary to appreciate VE-cadh disruption

â–º With confocal microscopy, which we have used for this study, the images in Fig. 5a were obtained by maximal magnification.

5. Fig 6 and Fig 7:

5-1) HE staining and collagen 4A staining could improve diabetic mouse phenotypage and the effect of midazolam.

â–º According to the comment, we performed additional experiments using Masson’s trichome and Sirius red staining to evaluate renal fibrosis. The results are displayed in new figures 7 and 8, and described in the Results (sections of 2.5 and 2.6; p9 to p10; the revised manuscript R1).

5-2) TEM images are very nice but weaker magnification is needed to appreciate the glomerulus as a whole and to visualize endothelial cells

â–º According to comment, we displayed TEM images with a lower magnification in Fig. 7a.

6. Discussion:

6-1) It has been shown that the TGase2 is involved in autophagy dependent clearance of ubiquitinated proteins. Moreover emerging data suggest that targeting autophagy is renoprotective during diabetic nephropathy. Thus, it will be interstesting to demontstrate in

this model if autophagy patway is modulated by TGase2 activation.

â–º It would be interesting to show the role of TGase2-associated autophagy in renal dysfunction. However, this study is out of the scope of our manuscript.  

6-2) the only marker used by authors to study adherns junctions is VE-cadh. however, it is necessary to complete this by ZO-1 staining, CD31 (PECAM1) staining and actin cytoskeleton.

â–º VE-cadherin is a key component of adherens junction and we reported adherens junction disassembly by VE-cadherin disruption in several papers, including Cardiovascular Research (2014), Diabetes (Ref. 9), and FASEB Journal (Ref. 10, 14). Thus, we think that VE-cadherin disruption is enough to suggest adherens junction disassembly for vascular leakage, and we do not think that it is unnecessary to demonstrate ZO-1, CD31, and actin cytoskeleton for adherens junction disassembly.

Round 2

Reviewer 3 Report

Thanks to the authors for correcting the manuscript. However, it is a pity that some points have not been taken into account:

Fig.4:Authors must show the consequences of TGase2 inhibition by siRNA on : i) VE-cadh staining, ii) ROS generation, iii) FITC-dextran

Fig5:

  • A bigger magnification of VE-cadh staining is necessary to appreciate VE-cadh disruption

The only marker used by authors to study adherns junctions is VE-cadh. however, it is necessary to complete this by ZO-1 staining, CD31 (PECAM1) staining and actin cytoskeleton.

Author Response

Response to reviewer’s comments R2

Ms. Ref. No.: IJMS-1478301R2

Title: Midazolam ameliorates hyperglycemia-induced glomerular endothelial dysfunction by inhibiting transglutaminase 2 in diabetes

[Reviewer #3]

1. Fig.4: Authors must show the consequences of TGase2 inhibition by siRNA on : i) VE-cadh staining, ii) ROS generation, iii) FITC-dextran

â–º According to the comments, we measured high glucose-induced VE-cadherin and endothelial cell permeability (FITC-detran) after transfecting HGECs with TGase2-specific siRNA. The results are described in the Results (p9, the revised manuscript R2), and displayed in Fig. 5e and f. However, we not think it is necessary to measure ROS generation because we showed that ROS generation is not affected by TGase2 activation in Fig. 4b and f. 

2. Fig5:

2-1. A bigger magnification of VE-cadh staining is necessary to appreciate VE-cadh disruption

â–º According to the comment, we showed magnified images for control and high glucose-treated cells in Fig. 5a.

2-2. The only marker used by authors to study adherns junctions is VE-cadh. However, it is necessary to complete this by ZO-1 staining, CD31 (PECAM1) staining, and actin cytoskeleton

â–º According to the comments, we stained actin filaments after treating endothelial cells with various inhibitors. The results are described in p8, and displayed in Fig. 5c. However, we do not think it is necessary to stain ZO because ZO is a component of tight junctions, but not that of adherens junctions. And we do not think it is necessary to stain CD31, a endothelial cell marker protein, because we performed the experiments in Fig. 5 in human glomerular microvascular endothelial cells, but not in kidney tissues.
